# Formulation and Evaluation of Chitosan/NaCl/Maltodextrin Microparticles as a Saltiness Enhancer: Study on the Optimization of Excipients for the Spray-Drying Process

**DOI:** 10.3390/polym13244302

**Published:** 2021-12-09

**Authors:** Shang-Ta Wang, Yi-Ying Lu, Min-Lang Tsai

**Affiliations:** Department of Food Science, National Taiwan Ocean University, Keelung 20224, Taiwan; awang@hughesbiotech.com (S.-T.W.); abc36633661@gmail.com (Y.-Y.L.)

**Keywords:** chitosan, spray drying, crystalline carbohydrates, hygroscopicity, saltiness perception, sodium reduction

## Abstract

Spray-dried chitosan/NaCl/maltodextrin microparticles have the potential to be used to enhance saltiness; however, its notable hygroscopicity results in handling and storage problems, thus limiting its application. In the present study, we attempted to introduce maltodextrin, microcrystalline cellulose (MCC), and waxy starch (WS) as excipients into the spray drying formulation of microparticles to reduce the cohesiveness and caking behavior and improve the yield simultaneously by ameliorating the moisture absorption tendency. The prepared microparticles showed a spherical appearance and had particle sizes ranging from 6.29 to 7.64 μm, while the sizes of the NaCl crystals embedded in the microparticles were 0.36 to 1.24 μm. The crystalline reflections of WS and MCC were retained in the microparticles after the spray-drying process. The handling properties were assessed to be acceptable. The formulation with only maltodextrin as the excipient showed a high moisture absorption rate of 2.83 g/100 g·h and a caking strength of 3.27 kg. The addition of MCC and WS significantly reduced the hygroscopic rate and caking strength. The spray-dried products provided better saltiness perception than native NaCl; as such, they may be promising for seasoning dry food products to achieve sodium intake reduction in the food industry.

## 1. Introduction

Sodium chloride (NaCl), also known as table salt, has been commonly used for food seasoning and preservation for thousands of years. It is made up of about 40% sodium, which plays a crucial role in the human body for maintaining biological functions, such as mediating the extracellular fluid volume, osmotic pressure, intestinal mineral absorption, and regulating the electrophysiological activity in muscles and nerve cells [1,2]. Apart from providing essential biological effects, excess sodium intake by the human body has been proven to lead to increasing risk of high blood pressure, cardiovascular disease, and stroke [3]. The World Health Organization (WHO) recommends a daily sodium intake (RDI) of less than 2 g (equivalent to 5 g salt/day); however, they reveal that most people around the world consume excess salt at an average of 9–12 g per day, which is about twice the recommended maximum level of intake [4]. Therefore, a member of WHO has agreed to take action at the population level to reduce dietary salt intake by a relative 30% by 2025. Additionally, the reduction of excess dietary salt has been identified as one of the most cost-effective means for improving population health worldwide [5,6,7]. However, decreasing salt content in foods may result in lower consumer acceptance, which could be implicated in the diminishing market shares of certain food products [8]. Thus, the sodium reduction strategy accompanied by maintaining the flavor and taste in food processing has received remarkable attention.

Various technologies have been introduced into the sodium reduction strategy. For example, the use of potassium salt, a typical sodium salt substitute, has been applied in the market for a long time, which could effectively overcome the high sodium content in processed foods [9]. Nevertheless, this kind of substance revealed off-tastes, including metallic, chemical, and bitter tastes, which are associated with potassium ions [10]. Their actual application in foods is, therefore, limited. Another approach for sodium reduction has drawn much attention, namely promoting the efficiency of sodium release. The rate of sodium release from the food matrix and the dissolution of salts from dry products have been demonstrated to be crucial for saltiness perception of foods. Thus, technologies such as reducing the size of salt crystals and modulating the texture of the food matrix have been recently applied for their potential as sodium reduction alternatives [11,12]. For example, Moncada et al. [13] used nano spray-drying technology to provide NaCl particles with extremely reduced particle size about 1.5 μm, and are capable to confer 25–50% sodium reduction capacity when used for seasoning of cheese crackers. Additionally, Cho et al. [14] performed NaCl/maltodextrin complexes with particle sizes at micrometer level by a spray-drying process, and it was effective to be dissolved faster and then saltiness was also rapidly recognized compared with native salt.

Chitosan, a linear polysaccharide derived from chitin, is composed of β-1,4 linked units of N-acetyl glucosamine and glucosamine and possesses a positive charge under weak acidic environments. It can be isolated and fabricated from bio-resources, including the exoskeletons of crustaceans, insects, mollusks, and fungi [15,16]. It attracts extensive attention in the biomedical field owing to its remarkable biological functions [17], and recently, the exploration of chitosan applications in food has expanded significantly [18]. Previously, we found that chitosan is capable of promoting the saltiness perception in food systems by incorporating the sodium chloride into microparticles to elevate the release of sodium ions [19,20]. Saltiness is mainly perceived from released free sodium ions rather than those bound with the food matrix. The free sodium ions in the oral cavity may bind to the taste bud receptor to provide the saltiness perception. However, when foods are consumed, generally a high proportion of sodium may remain in the food matrix in its bound form even after being swallowed. Consequently, a significant amount of sodium may be absorbed into the body without being perceived, which means the sodium intake would be much higher than the quantities that were actually needed for adequate saltiness perception in these circumstances [21]. It is reported that the existence of negatively charged molecules in the food matrix, such as milk protein, soy protein, xanthan gum, and κ-carrageenan, would diminish the saltiness perception due to the increased electrostatic interactions between the sodium ions and the food matrix [22]. In contrast, chitosan possesses positively charged functional groups, which may interact with negatively charged groups through ionic interactions. Therefore, it would be capable of helping to release free sodium ions and further enhance the saltiness perception [19,20].

We previously fabricated chitosan/NaCl/maltodextrin microparticles using a spray-drying process [19] and found that particles made of chitosan with a 78.3% degree of deacetylation (DD), owing to its significant positive charge characteristic, are capable of achieving a quarter sodium reduction without a reduction in flavor by using this product as a salt replacement. However, the average yield of the microparticles is about 30%, which is unacceptable for industrial applications. Therefore, we further added maltodextrin as a basic excipient in the formulation [20], which resulted in a ~2 times higher yield than the original formulation. The saltiness-enhancing capabilities were also comparable to the original. However, there are still drawbacks in the products from the modified formulation, namely, the high hygroscopicity and caking property as a result of the high moisture absorption tendency of the amorphous maltodextrin molecules. Accordingly, in the present study, we aimed to optimize the product behavior by introducing crystalline carbohydrates, including microcrystalline cellulose (MCC) and waxy starch (WS), as excipients into the formulation. The yield, density, size, shape, flowability, dissolution rate, and salinity of the chitosan/NaCl/maltodextrin microparticles with various crystalline carbohydrate content were investigated. The hygroscopicity was also assessed as an indicator to evaluate the potential for further applications.

## 2. Materials and Methods

### 2.1. Materials

Chitosan was from Charming & Beauty Co. (Taipei, Taiwan). Salt (>99% NaCl) was from Taiyen Industrial Co., Ltd. (Tainan, Taiwan). Maltodextrin (dextrose equivalent = 10) was from Gemfont Co. (Taipei, Taiwan). High amylopectin waxy maize starch (N-200) was from Roquette Frères (Lestrem, France). Microcrystalline cellulose (Comprecel^®^ M101LD, bulk density = 0.14–0.24 g/mL) was from Mingtai Chemical Co. Ltd. (Taoyuan, Taiwan). 

### 2.2. Chemicals

Acetic acid and sodium azide was from Merck & Co. (Darmstadt, Germany). Potassium bromide and sodium sulfate was from Riedel-de haën (Seelze, Germany). Sodium acetate, phenolphthalein, potassium chromate, silver nitrate, and sodium hydroxide were from Sigma-Aldrich Corp. (St. Louis, MO, USA).

### 2.3. Degree of Deacetylation (DD) and Molecular Weight of Chitosan

The DD of the chitosan was determined using Fourier transform infrared spectroscopy (FTIR) [23]. In brief, the chitosan powder was mixed with KBr at a ratio of 1:100 and then dried at 60 °C for 3 days before being pressed into a pellet form. The absorbance of amide 1 (1655 cm^−^^1^) and the hydroxyl band (3450 cm^−^^1^) was measured using an FTIR spectrometer (Bio-Rad FTS-155, Hercules, CA, USA). Triplicate measurements were averaged and applied to calculate the DD by using the following equation:DD (%) = 100 − (A_1655_/A_3450_) × 115(1)

The molecular weight of chitosan was assessed by an established size exclusion high-performance liquid chromatography method with minor modification [24]. A column (7.8 mm × 30 cm) packed with TSK gel G4000 PW_XL_ and G5000 PW_XL_ (Tosoh Co., Ltd., Tokyo, Japan) was used. The mobile phase consisted of 0.2 M acetic acid/0.1 M sodium acetate and 0.008 M sodium azide. The sample at a concentration of 0.5% (*w*/*v*) was loaded and eluted with a flow rate of 0.6 mL/min by a Shimadzu LC-10AT pump (Tokyo, Japan). The elute peak was detected by a Shodex RI-71 RI detector (Tokyo, Japan). Pullulans with determined molecular weight were used as standards to prepare the calibration curve, which was used to calculate the molecular weight of chitosan. 

### 2.4. Preparation of Chitosan/NaCl/Maltodextrin Microparticles by Spray Drying

The formulations were divided into seven batches prepared with different ratios of NaCl and suitably chosen polymers, as depicted in Table 1. CNM indicates the formulation with only maltodextrin as the excipient. MCC indicates the formulations with maltodextrin and microcrystalline cellulose as the excipients. WS indicates the formulations with maltodextrin and waxy maize starch as the excipients. The mixtures were dissolved in a 0.4% (*v*/*v*) acetic acid solution. Spray drying was carried out by using a lab-scaled spray dryer (EYELA SD-1000, Tokyo Rikakikai Co., Ltd., Tokyo, Japan) with a two-way nozzle. The rotating speed of the nozzle was 400 rpm, the feeding flow rate of the solution was 400 mL/h, and the inlet and outlet temperatures were 150 and 95 °C, respectively. The aeration rate of hot air was 0.7 m^3^/min, and the system pressure was 90 kPa [20]. The spray-dried product was collected and weighed. The yield was calculated as follows:Yield (%) = (Sample weight/Solute weight) × 100%(2)

### 2.5. X-ray Diffraction

Crystallinities of chitosan microparticles were analyzed by an X-ray diffractometer (D2 PHASER, X’Pert Pro MPD, Bruker, AXS Inc., Madison, WI, USA). The spray-dried products were compressed and mounted on the sample tray. The analysis was carried out with an anode current of 20 mA and an accelerating voltage of 40 kV. Samples were exposed to CuKα radiation at diffraction angles (2θ) from 5° to 60°, and the counting time was 0.5 s at each angle step (0.1°) [23]. 

### 2.6. Scanning Electron Microscopy (SEM)

The spray-dried microparticles were securely attached onto the aluminum ingots by carbon gel. Subsequently, a gold-plating machine (ion-sputter, HitachiE-1010; Tokyo, Japan) was used to plate 2-nm-thick platinum films onto the samples. The morphologies of the samples were observed using SEM (Hitachi SEM S-3400N; Tokyo, Japan) [25], and their particle sizes (n = 50) were measured and calculated using Image J software (NIH, USA).

### 2.7. Moisture, Sodium Chloride, and Acetic Acid Content of Microparticles

The moisture content (%) of the spray dried microparticles was determined by infrared drying using a moisture analyzer (Denver Instrument IR-35, Goettingen, Germany). Five-hundred milligrams of sample was placed on the aluminum plate at 105 °C for 60 min, and the weight change was monitored for calculating the moisture content. 

For determining the NaCl content, 1 g of the sample was dissolved in 200 mL of water. Then, a 10 mL aliquot was transferred to an Erlenmeyer flask and mixed with 1 mL of the 5% K_2_CrO_4_ titration indicator. The sample was titrated with a 0.1 N AgNO_3_ solution until an orange–brown color was maintained for 30 s, and the titration volume was recorded. The NaCl content was determined as follows:NaCl (%) = (T × 0.00585 × D)/S × 100%(3)
where S is the sample weight (g), T is the titration volume (mL), D is the dilution factor (total volume/titration volume), and 0.00585 g NaCl is equivalent to 1 mL of a standard 0.1 N AgNO_3_ solution.

The acetic acid content of the microparticles was determined using acid–base titration. In brief, 1 g of the sample was suspended in 200 mL water and stirred. Then, a 10-mL aliquot was transferred to an Erlenmeyer flask. Three drops of a phenolphthalein indicator were added. Subsequently, a 0.1 N NaOH solution was used for titration until the solution turned pink. The titration volume was then recorded, and the acetic acid content was calculated as follows:Acetic acid content (%) = (T × F × D × E)/S × 100%(4)
where S is the sample weight (g), T is the titration volume (mL), F is the factor for a 0.01 N NaOH solution, D is the dilution factor (total volume/titration volume), and E is the equivalent weight of acetic and lactic acid in 1 mL of a 0.1 N NaOH solution, which was 0.0060 and 0.0090 g, respectively.

### 2.8. Densities and Flowabilities of Microparticles

Bulk and tapped densities were measured by using a 10 mL graduated cylinder [20]. The spray-dried sample was poured into the cylinder and tapped mechanically 100 times. Then, the tapped volume was addressed, and the bulk density and tapped density were calculated. Each experiment was performed in a triplicate manner. 

In addition, the Carr index (CI) and Hausner ratio (HR) were calculated for evaluating the flowabilities as follows [26]:(5)$$\mathrm{CI}=\frac{\mathrm{Tapped}\;\mathrm{density}-\mathrm{Bulk}\;\mathrm{density}}{\mathrm{Tapped}\;\mathrm{density}}\times 100$$
(6)$$\mathrm{HR}=\frac{\mathrm{Tapped}\;\mathrm{density}}{\mathrm{Bulk}\;\mathrm{density}}$$

### 2.9. Hygroscopicity Assessment 

The hygroscopicity of the samples was determined by using the Conway dish method. In brief, the microparticles were placed in petri dishes and allowed to settle inside the closed containers containing a saturated Na_2_SO_4_ solution (75% RH). The samples were aliquoted after 12, 24, 36, 48, and 72 h, respectively, and weighed to measure the amount of water that was absorbed from the environment. The results were presented as grams of water absorbed per 100 g of dry matter (g/100 g). The hygroscopicity of the samples over time was plotted, and regression calculations were performed to determine the slopes of the curves, which represented the hygroscopic rates (g/100 g·h) of the samples [19].

### 2.10. Determination of Caking Strength

One gram of powder was weighed into the plastic bottle and taped gently to make a flat layer. Then, the powders were compacted at 0.1 mm/s with a compression strength of 1 kg to reach a compression volume ratio of 20% by using a TA-XT2i Texture Analyzer (Stable Micro System, Godalming, UK). After that, the compacted cake together with the plastic bottle was stored under a predetermined environment at a controlled temperature of 25 °C and 43.2% RH for 10 days. At the end of the storage, the powder cake plug was gently taken out of the bottle. A compression test for analyzing the hardness of the powder cake plug was carried out by using a 6 mm probe attached to the TA-XT plus Texture Analyzer [27]. The required strength needed to break the powder cake plug was determined to be the caking strength. 

### 2.11. Dissolution Rate in Artificial Saliva

The artificial saliva (SAGF) was prepared according to the formula previously reported [28]. Five hundred milligrams of the microparticle samples were added to 50 mL of SAGF at 25 °C. The dissolution of salts was determined by monitoring the conductivity of the solution using a conductivity meter (SC-2300, Suntex, Taipei, Taiwan) at 60 rpm of magnetic stirring within 20 min. Data collection for the dissolution rates began when the salt sample was added to the SAGF solution. The conductivity measurements were conducted at 2 s intervals until the maximum conductivity was reached. It was assumed that at the maximum conductivity, the sample had achieved total dissolution. The conductivity at each time point was plotted to generate the dissolution curves. The times for reaching 20 and 50% dissolution (T_20%_, T_50%_) were calculated.

### 2.12. Sensory Evaluation

To alleviate the particle size variation of each group, which may lead to misjudgments of the salinity potential, the general NaCl was ground and passed through a standard sieve. Besides the general NaCl, particles with a size of 53–75 μm were collected and used in the salinity evaluation as well. The experimental protocol was approved by the Institutional Behavior and Social Science Committee of National Taiwan University under the project number NTU-REC: 201705EM038. Postgraduates of the Department of Food Science at National Taiwan Ocean University (Keelung, Taiwan) were recruited to participate in a hedonic taste test. The panel consisted of 50 untrained volunteer subjects (33 females and 17 males, 21–28 years old). For the chitosan/NaCl/maltodextrin microparticles, 20 g of corn kernels was popped and 1 g of NaCl or chitosan/NaCl/maltodextrin microparticles was spread evenly on the surface. The salinity of the general NaCl group was given a score of 0 and set as the standard for comparison with the other groups. Groups that were scored as less salty than that of the NaCl group were given a negative score of as low as −5, whereas those that were scored as saltier were given a positive score of as high as 5. The participants were asked to rate and compare only the differences in the salinity of the samples and to ignore subjective factors such as personal preference, acceptability, and other tastes [19].

### 2.13. Statistical Analysis

One-way analysis of variance (ANOVA) was performed using SPSS version 22 for Windows (IBM Corp., Armonk, NY, USA) to analyze the data. Multiple comparisons among means were performed using a variation of Duncan’s new multiple range test for the ANOVA results with significant differences. The significance level was set at *p* < 0.05.

## 3. Results

### 3.1. Preparation and Characterization of Chitosan and Its Microparticles

The DD and average molecular weight of chitosan used in this study were determined to be 76.59 ± 1.44% and 87.57 ± 1.22 kDa, respectively. 

#### 3.1.1. The Yields of Chitosan/NaCl/Maltodextrin Microparticles

The yields of chitosan/NaCl/maltodextrin microparticles prepared with each formulation were listed in Table 2, which were ranged from 39.29 to 55.65%. Formulation without addition of crystalline carbohydrates showed highest yield among all groups while other groups revealed relative low yields, about 40%. However, this refined process contributes to significant improvement in yields compared to a previous report [19]. Yields of formulations with MCC were higher than that of WS groups, but the discrepancies are negligible. 

#### 3.1.2. Moisture, NaCl, and Acetic Acid Content of Chitosan/NaCl/Maltodextrin Microparticles

The moisture, NaCl, and acetic acid content of the spray-dried products are shown in Table 2. The moisture content ranged from 5.04–7.37%. The addition of crystalline carbohydrates significantly reduced the moisture content of the microparticles; however, the excess content of crystalline carbohydrates resulted in elevated moisture content, which could be observed in the MCC20 and WS20 groups. The moisture content of the chitosan/NaCl/maltodextrin microparticles in the present study was comparable to those composed of only chitosan and NaCl [19] and significantly improved from those with the addition of maltodextrin from a previous report [20]. This indicates that the crystalline carbohydrates may offset the moisture absorption tendency of maltodextrin in the microparticle system. For the NaCl content, it occupied about a quarter of the total weights of the spray-dried microparticles with various excipients, which ranged from 25.35 to 28.86%. The acetic acid content ranged from 1.60 to 2.40%, which showed no significant difference between the groups.

#### 3.1.3. Morphology and Particle Size

The morphology of the chitosan/NaCl/maltodextrin microparticles was studied via SEM. Figure 1 shows the spherical shape and various droplet sizes of the particles. In the micrograms, the embedded NaCl crystals with varied sizes were observed all over the particle surface. Ruptured structures were observed in the microparticles without the addition of crystalline carbohydrates (Figure 1A). Microparticles with MCC as the excipient showed surface indentations in their appearance, representing the shrinkage formations of these particles (Figure 1B–D). 

The average particle size of the microparticles and NaCl crystals was calculated from the micrographs of each group by using Image J software (Figure 1A–G). The particle size of whole chitosan/NaCl/maltodextrin microparticles ranged from 6.29 to 7.24 μm (Table 3). The groups with the smallest and largest size were MCC10 and WS10, respectively. For the size of the embedded NaCl crystals, the addition of crystalline carbohydrates was found to be associated with an increased particle size. The size of the NaCl crystals in the CMN group was 0.36 μm, whereas that of the other groups was above 0.43 μm (Table 3). The WS groups possessed a significantly larger NaCl crystal particle size than the MCC groups. 

#### 3.1.4. X-ray Diffraction

X-ray diffraction patterns of the tested samples are shown in Figure 2. Four strong peaks in the NaCl diffractogram were found at 2θ = 27, 31, 45, and 57° (Figure 2A). Maltodextrin and chitosan revealed typical crystalline reflections corresponding to 2θ = 18 and 20° in the diffractogram, respectively (Figure 2B,C) [29,30]. CMN microparticles possessed significant typical crystalline reflections of NaCl and minor typical crystalline reflections of chitosan (Figure 2D). However, the crystalline reflection of maltodextrin was not found in the CMN diffractogram, suggesting that the crystalline region of maltodextrin materials was fully disrupted through the dissolving and spray-drying process. Moreover, the typical crystalline reflections of chitosan, MCC, and WS were retained in the diffractogram of their corresponding formulations as minor reflection peaks (Figure 2F–H,J–L), indicating that the components were uniformly assembled into microparticles.

### 3.2. Densities and Flowability of Chitosan/NaCl/Maltodextrin Microparticles

The bulk density and tapped density are both crucial characteristics of spray-dried powder due to functional and cost-effective reasons [31]. In the present study, bulk densities of the microparticles varied from 0.28 to 0.34 g/cm^3^, and the tap densities ranged from 0.34 to 0.40 g/cm^3^ (Table 4). On the other hand, flowability indices, including the Hausner ratio and Carr index, represent the flow behavior of powders under temperature and humidity, which is important in product handling and utilization. The Carr index classified the excellent, good, fair, and passable quality powder at 0–10, 11–15, 16–20, and 21–25%, respectively, whereas the Hausner ratio ranges were 1.00–1.11, 1.12–1.18, 1.19–1.25, and 1.26–1.34, accordingly [32]. The spray-dried microparticle powder had almost all similar flow characteristics for all treatments in the present study and were considered as “passable” and “fair” powders by their Hausner ratio and Carr index given in Table 4. The addition of CMC helped the chitosan/NaCl/maltodextrin microparticles fall into the “fair” category, whereas the WS groups almost fall into the “passable” category accordingly. The spray-dried products with MCC as the excipient show better handling properties than the other tested groups.

### 3.3. Hygroscopicity and Caking Strength

The water absorption capacities translated from the moisture absorption–time curves (data not shown) within 36 and 72 h of the chitosan/NaCl microparticles are shown in Table 5. The values ranged from 2.32–4.45 g/100 g·h. Comparing to the results from a previous report [19], the hygroscopicity of the CNM group was significantly higher. However, with the addition of crystalline carbohydrates, the progression of the moisture absorption of the microparticles was significantly reduced to comparable values to previous [19]. The addition of more MCC and WS in the formulation resulted in a more proper hygroscopic rate for the spray dried products.

Caking is a complex chemical and physical process, while the strength involves coupled intrinsic cohesion strength caused by heat and moisture transfer in the products [33]. The experimental caking strengths of the samples composed of different formulations in the present study are listed in Table 5, which showed identical trends as that of the results from the hygroscopic rate. Among all groups, WCC20 reduced almost a quarter of the caking strength compared to the CNM group.

### 3.4. Dissolution Rates of NaCl in the Microparticles

In the present study, we investigated the accessibilities of NaCl from the microparticle matrices by using an artificial saliva dissolution model with an electrical conductivity meter. The conductivity–time curves are shown in Figure 3. Among all groups, MCC20 revealed a rapid elevation of conductivity at the early stage, whereas the dissolution rate of MCC10 was retarded. However, all experimental groups showed comparable NaCl release at the end point of the experiment. We calculated the time required for a quantity of NaCl to be liberated to half of its total content as T_50%_ from the regression curves. The T_50%_ of CNM, MCC10, MCC15, MCC20, WS10, WS15, and WS20 were 133, 165, 132, 68, 109, 145, and 88 s, respectively. Accordingly, the addition of crystalline carbohydrates would not impede the release of NaCl into the saliva except for the MCC10 group, which showed a minor elevation in the T_50%_ value. For reference, we also conducted the dissolution study of native NaCl; the dissolution was rapid and resulted in a T_50%_ value of 27 s.

### 3.5. Sensory Evaluation

In the present study, we aimed to improve the handling properties and hygroscopicity of chitosan/NaCl/maltodextrin microparticles. Considering the results from the aforementioned experiments, we selected the MCC15, MCC20, and WS20 groups for further sensory evaluation. However, the MCC10 group was excluded due to the lower dissolution rate, which was speculated to negatively affect the saltiness perception of the microparticles. For comparison, the sensory evaluation of the CNM group, native, and grounded NaCl was also conducted. The outcomes are shown in Figure 4. The CMN, MCC15, MCC20, and WS20 groups provided better saltiness perception than native NaCl. This could be associated with the numerous nano- to micrometer-level NaCl crystals on the surface of the spray-dried chitosan/NaCl/maltodextrin microparticles (Figure 1). The observed tiny NaCl particles may be able to provide a relatively large surface area, resulting in a stronger saltiness taste.

## 4. Discussion

In the present study, chitosan/NaCl/maltodextrin microparticles were prepared based on designed formulations by using crystalline carbohydrates as excipients. Due to the need of an acid environment for dissolving chitosan, the residual acid in the microparticles is inevitable. According to a previous report [20], acetic acid—a typical volatile acid, which may mostly be evaporated during the preliminary stage of the spray-drying process before forming the aggregates and may provide better yields than that composed of other organic acids—was used in this study. On the other hand, the temperature of the spray dryer is an important factor affecting the properties of resulted microparticles. In the present study, we have tried various process conditions, and found that the microparticles with best handling properties could be obtained at 150 °C of inlet temperature. The products obtained with inlet temperatures higher than 150 °C performed glutinous behavior of products. Moreover, via the differential scanning calorimetry analysis, we found that there are phase transitions occurred within the microparticles at above 185 °C (Appendix A), representing the rupture of crystalline structure, and may result in unacceptable handling property. Therefore, the process temperature should be limited to 185 °C of proposed formulations in this study.

In the morphology study, we found that without the existence of crystalline carbohydrates, the microparticles would be ruptured and form hollow structures, which may absorb more moisture in certain extent as the result in Table 2. Adding chitosan in the spray-drying formulation resulted in rough particle surface [34], and are therefore capable of retaining NaCl crystals. Additionally, based on the results from X-ray diffraction (Figure 2), only minor peaks of typical chitosan crystalline reflections were found in the diffractograms of chitosan/NaCl/maltodextrin microparticles, indicating that the crystalline structures of chitosan no longer remained when forming the microparticles. In this condition, the chitosan molecules are assumed to be amorphous and are capable of providing more positively charged groups to help enhance the saltiness perception than the chitin nanomaterials we previously fabricated [23]. Through the SEM observation, we also found that the embedded NaCl crystals of the WS groups possessed a larger particle size than the others. This may be the consequence of gelatinization of waxy starch within the spray-drying process, which led to the internal moisture of microparticles diffusing to the outer layer. This further caused the merging of NaCl crystals into larger particles [30]. This may be one of the factors that lead to a lower saltiness perception of the WS groups than the CMC groups (Figure 4). On the other hand, the particle size of obtained microparticles in the present study are almost six times lower than that of previous reports, in which the spray dried particles were composed of maltodextrin and soy sauce [27].

In terms of the handling properties, the spray-dried microparticles in the present study all fall into the categories of “fair” and “passable” powders. This may be ascribed to their small particle size, leading to a relatively large surface area per unit mass of powder, which provides more contact surface area between the powder particles available for cohesive forces to restrict flowability [35]. The formulation with organic acid of spray-dried powder was also reported to have a relatively low flowability [36]. Even though the flowability in the present study was not quite optimal, the obtained microparticles could have significantly lower moisture contents compared to several maltodextrin-based spray-dried food powders in previous report [37,38]. Despite this, based on the quality of the powder, the range from excellent to passable is acceptable for spray-dried powders in terms of their handling properties [31]. On the other hand, the storage properties were basically influenced by the hygroscopicity and caking strength of the spray-dried powder. In the present study, the hygroscopic rates were significantly reduced along with the addition of crystalline carbohydrates. MCC was reported as a good carrier material to reduce the hygroscopicity of maltodextrin, which would overcome the stickiness and stability problem to a certain degree during storage [39]. In this study, the combination of the carrier materials resulted in a partial crystalline structure remaining in the spray-dried chitosan/NaCl/maltodextrin powder, therefore resulting in a less sticky product. Aside from the MCC, WS is also capable of reducing the amorphous behavior of the spray-dried powder [40]. According to our results, the microparticles composed of WS possessed the retained crystalline structures of WS (Figure 2), and the hygroscopic rates were therefore retarded (Table 5), especially in the WS20 group, which provided the lowest hydroscopic tendency among all tested groups. 

Adding the crystalline carbohydrates also helps reduce the caking strength of the chitosan/NaCl/maltodextrin powder. According to a previous report, a spray-dried soy sauce powder with 20 and 40% maltodextrin (DE = 10) possessed about 2 and 7 kg of caking strength at a relative humidity of 43%, respectively [27]. In our study, the tested microparticles with about 50–60% maltodextrin (DE = 10) possessed caking strengths ranging from 2.57 to 4.42 kg at identical conditions to the previous study. The diminished caking strength may be due to the crystalline behaviors contributed by MCC and WS in the microparticle architecture. However, the powder solubility has been reported to be negatively affected by the addition of MCC and WS [27], and moreover, the saltiness-enhancing mechanism of chitosan-related materials are assumed to be the positive charges from protonated amino groups of these chitosans, which may form ionic interactions with the negatively charged ions in the food system and facilitate the release of free sodium ions in the oral cavity [18,19]. The assumption has been confirmed by comparing the saltiness-enhancing capacities of chitosan/NaCl/maltodextrin in solution and in dry food systems, which previously suggested that the saltiness perception was elevated in dry food but not the solution system [19]. Therefore, the microparticles fabricated in this study may be considered suitable for application onto salted dry foods such as potato chips and walnuts rather than liquid types of foods. 

The release of NaCl from microparticles would be a crucial process for the spray-dried products to provide saltiness. It was assumed that the existence of crystalline carbohydrates in the system may facilitate the liberation of NaCl owing to the higher crystalline behavior of the microparticles. Unexpectedly, the dissolution rates of almost all groups with crystalline carbohydrates were retarded at the initial period of the experiment. It may be because the negatively charged groups of these polysaccharides shrouded the positive-charged groups from chitosan molecules on the surface [41]. Therefore, a certain quantity of sodium ions was retained. In the sensory evaluation, we found that all tested microparticles were capable of providing better saltiness perceptions than native NaCl. On the basis of the NaCl content of each group, the usage of MCC15, MCC20, and WS20 may be capable of reducing the sodium intake by at least 71, 73, and 75% without the impairment of the saltiness of the food, respectively, which may be a remarkable outcome that is superior to that of chitin-based materials previously reported [23,42,43].

The use of saltiness enhancer is a typical strategy for sodium reduction. In general, these perception enhancement agents are mostly based on small molecules or bio-material extracts such as monosodium glutamate, and yeast extract [44,45], rather than the polymers. Only a few reports on the saltiness-enhancing capacity of polymers have appeared in the literature, and in almost all cases, they are anionic polymer-based material [14,27]. In fact, the existence of polymers in food systems were reported not only invalid but impaired the saltiness. Some cellulose-based food thickeners such as carboxymethylcellulose and hydroxypropyl-methylcellulose, were found to cause the reduction in saltiness perception in foods [46]. By this study, however, we found that the cellulose-based material MCC may help improving the handling properties of our spray-dried products without negatively affecting the saltiness, indicating an extended application of these polymers. Additionally, through the present study, we addressed a novel application of cationic biopolymer chitosan and extended the feasibility of the composed microparticles in industrial applications.

## 5. Conclusions

In conclusion, chitosan/NaCl/maltodextrin microparticles with maltodextrin, MCC, and WS as excipients in various proportions were prepared by spray-drying technology. The addition of MCC and WS significantly reduced the hygroscopic rate and caking strength of the microparticles. Among all formulations, MCC15 possessed acceptable handling properties and a relatively lower hygroscopic rate and caking strength with considerable saltiness enhancement, simultaneously. This could be considered an optimal formulation for further investigation. Furthermore, even though the yield and handling properties are acceptable, refinement of the process conditions, such as amelioration of the inlet temperature of spray drying, is needed.

## Figures and Tables

**Figure 1 polymers-13-04302-f001:**
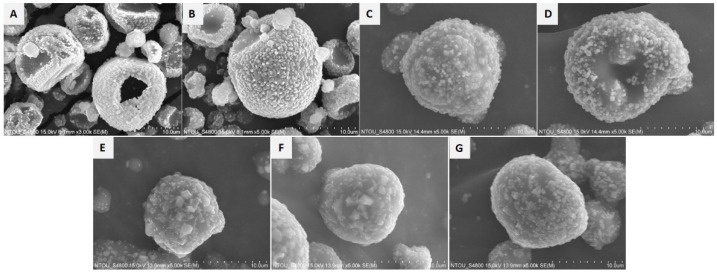
Scanning electron microscopy (SEM) micrographs of spray-dried chitosan/NaCl/maltodextrin microparticles with different formulations. (**A**) CNM; (**B**) MCC10; (**C**) MCC15; (**D**) MCC20; (**E**) WS10; (**F**) WS15; (**G**) WS20. The abbreviations and compositions of each formulation are described in Table 1.

**Figure 2 polymers-13-04302-f002:**
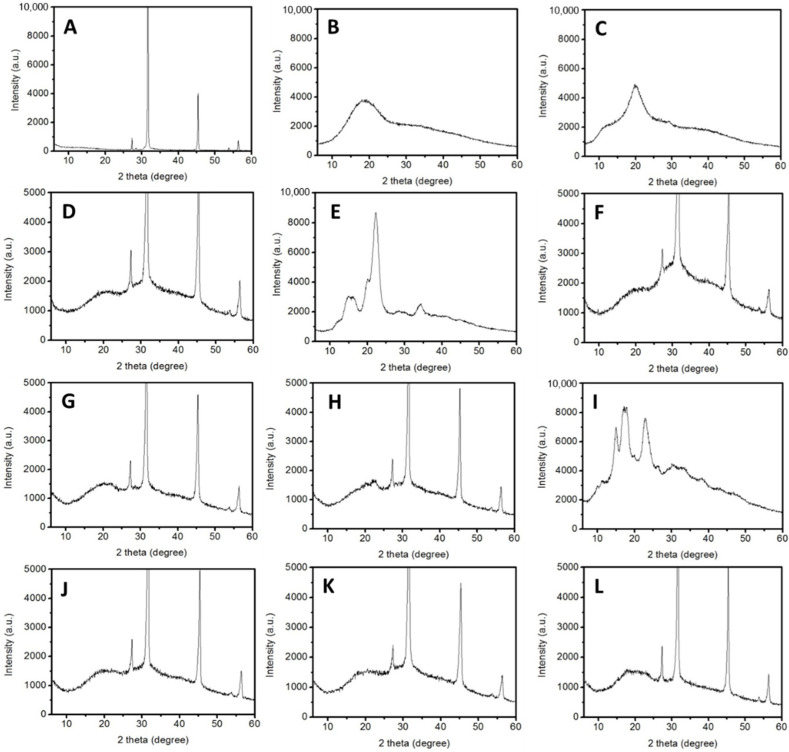
X-ray diffraction patterns of (**A**) NaCl; (**B**) maltodextrin; (**C**) chitosan; (**D**) CMN; (**E**) MCC alone; (**F**) MCC10; (**G**) MCC15; (**H**) MCC20; (**I**) WS alone; (**J**) WS10; (**K**) WS15; (**L**) WS 20. The abbreviations and compositions of each formulation are the same as in Table 1.

**Figure 3 polymers-13-04302-f003:**
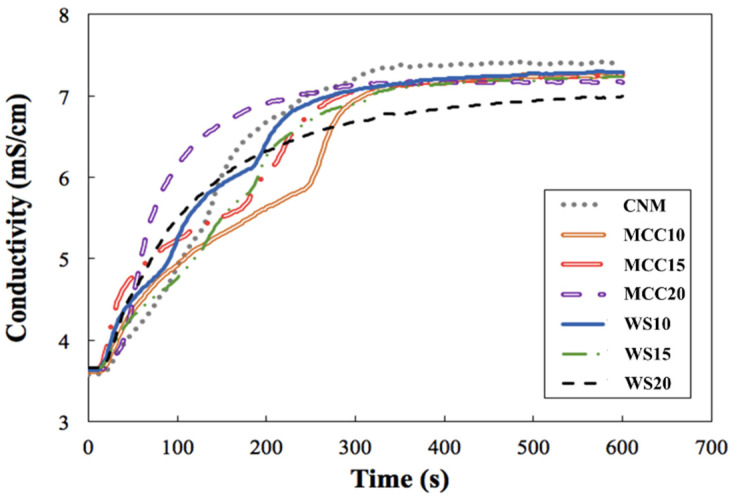
Conductivity–time curves of chitosan/NaCl/maltodextrin microparticles. The abbreviations and compositions of each formulation are the same as in Table 1.

**Figure 4 polymers-13-04302-f004:**
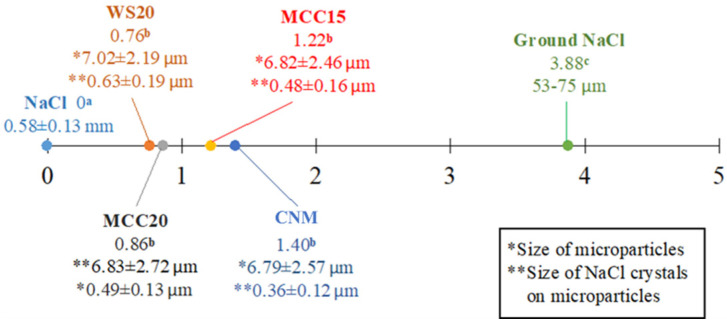
The sensory evaluation results of ground NaCl and each experimental group in the dry food system. The abbreviations and compositions of each formulation are the same as in Table 1. ^a–c^ Values with different superscripts indicate significant differences according to Duncan’s new multiple range test (*p* < 0.05).

**Table 1 polymers-13-04302-t001:** The composition (%, *w*/*w*) of each formulation.

	NaCl	Chitosan	Maltodextrin	Microcrystalline Cellulose	Waxy Maize Starch
CNM	30.6	4.3	65.1	0	0
MCC10	27.2	3.8	58.0	11.0	0
MCC15	25.5	3.7	54.5	16.3	0
MCC20	23.9	3.5	51.0	21.6	0
WS10	27.2	3.8	58.0	0	11.0
WS15	25.5	3.7	54.5	0	16.3
WS20	23.9	3.5	51.0	0	21.6

CNM indicates the formulation with only maltodextrin as the excipient. MCC indicates the formulations with maltodextrin and microcrystalline cellulose as the excipients. WS indicates the formulations with maltodextrin and waxy maize starch as the excipients.

**Table 2 polymers-13-04302-t002:** The yields, moisture, acetic acid, and NaCl content of chitosan/NaCl/maltodextrin microparticles.

	Yield (%)	Moisture Content (%)	NaCl Content (%)	Acetic Acid Content (%)
CNM	55.65	7.37 ± 0.68 ^c^	27.69 ± 0.55 ^bc^	2.00 ± 0.28 ^a^
MCC10	43.88	5.18 ± 0.90 ^a^	27.69 ± 0.55 ^bc^	2.40 ± 0.49 ^a^
MCC15	43.60	5.04 ± 0.91 ^a^	28.86 ± 0.73 ^c^	2.20 ± 0.28 ^a^
MCC20	44.80	6.12 ± 0.30 ^abc^	26.72 ± 0.28 ^ab^	2.10 ± 0.24 ^a^
WS10	40.83	5.82 ± 0.32 ^ab^	28.47 ± 1.46 ^c^	1.60 ± 0.28 ^a^
WS15	44.21	5.35 ± 0.47 ^ab^	27.69 ± 0.55 ^bc^	1.90 ± 0.14 ^a^
WS20	39.29	6.68 ± 0.24 ^bc^	25.35 ± 0.28 ^a^	1.60 ± 0.28 ^a^

The abbreviations of the groups are the same as in Table 1. The data except the yield values are expressed as mean values ± S.D. (n = 3). ^a–c^ Values with different superscripts in the same column indicate significant differences according to Duncan’s new multiple range test (*p* < 0.05).

**Table 3 polymers-13-04302-t003:** The average particle size of the whole spray dried microparticles and embedded NaCl crystals.

	Microparticles (μm)	NaCl Crystals (μm)
CNM	6.79 ± 2.57 ^a^	0.36 ± 0.12 ^a^
MCC10	6.29 ± 1.97 ^a^	0.43 ± 0.11 ^ab^
MCC15	6.94 ± 2.75 ^a^	0.48 ± 0.16 ^ab^
MCC20	7.03 ± 3.37 ^a^	0.49 ± 0.13 ^b^
WS10	7.64 ± 2.20 ^b^	1.24 ± 0.36 ^d^
WS15	6.76 ± 2.58 ^a^	0.66 ± 0.25 ^c^
WS20	7.02 ± 2.19 ^ab^	0.63 ± 0.19 ^c^

The abbreviations and compositions of each formulation are the same as in Table 1. Data were expressed as mean values ± S.D. (n ≥ 80). ^a–d^ Values with different superscripts in the same column indicate significant differences according to Duncan’s new multiple range test (*p* < 0.05).

**Table 4 polymers-13-04302-t004:** The bulk density, tapped density, and flowability indices of chitosan/NaCl/maltodextrin microparticles.

	Bulk Density (g/cm^3^)	Tapped Density (g/cm^3^)	Flowability	
			Hausner Ratio	Carr Index
CNM	0.29 ± 0.02 ^b^	0.34 ± 0.01 ^a^	1.33 ± 0.03 ^ab^	24.68 ± 1.49 ^ab^
MCC10	0.28 ± 0.00 ^b^	0.37 ± 0.01 ^ab^	1.19 ± 0.04 ^a^	15.60 ± 2.77 ^a^
MCC15	0.29 ± 0.02 ^b^	0.36 ± 0.02 ^ab^	1.23 ± 0.00 ^ab^	18.90 ± 0.23 ^a^
MCC20	0.32 ± 0.02 ^c^	0.40 ± 0.01 ^bc^	1.25 ± 0.08 ^ab^	19.90 ± 5.27 ^ab^
WS10	0.24 ± 0.01 ^a^	0.35 ± 0.05 ^a^	1.43 ± 0.18 ^b^	28.87 ± 8.71 ^b^
WS15	0.28 ± 0.01 ^b^	0.38 ± 0.01 ^abc^	1.37 ± 0.06 ^ab^	27.04 ± 3.25 ^b^
WS20	0.34 ± 0.00 ^c^	0.42 ± 0.00 ^c^	1.24 ± 0.01 ^ab^	19.50 ± 0.70 ^b^

The abbreviations and compositions of each formulation are the same as in Table 1. Data were expressed as mean values ± S.D. (n = 3). ^a^^–c^ Values with different superscripts in the same column indicate significant differences according to Duncan’s new multiple range test (*p* < 0.05).

**Table 5 polymers-13-04302-t005:** The hygroscopic rate and caking strength of chitosan/NaCl/maltodextrin microparticles.

	Hygroscopic Rate 0–36 h (g/100 g·h)	Hygroscopic Rate 0–72 h (g/100 g·h)	Caking Strength (kg)
CNM	4.45 ± 0.14 ^c^	2.83 ± 0.10 ^c^	3.27 ± 0.21 ^ab^
MCC10	3.80 ± 0.55 ^bc^	2.52 ± 0.17 ^b^	2.91 ± 0.21 ^bc^
MCC15	3.94 ± 0.10 ^ab^	2.33 ± 0.16 ^a^	2.97 ± 0.29 ^ab^
MCC20	3.74 ± 0.07 ^a^	2.33 ± 0.05 ^a^	2.57 ± 0.26 ^a^
WS10	4.19 ± 0.10 ^bc^	2.59 ± 0.05 ^b^	4.42 ± 0.38 ^c^
WS15	4.09 ± 0.11 ^b^	2.56 ± 0.11 ^b^	3.79 ± 0.20 ^bc^
WS20	3.69 ± 0.28 ^a^	2.32 ± 0.12 ^a^	3.26 ± 0.28 ^ab^

The abbreviations and compositions of each formulation are the same as in Table 1. Data from hygroscopic rate (n = 3) and caking strength (n = 2) were expressed as mean values ± S.D. ^a^^–c^ Values with different superscripts in the same column indicate significant differences according to Duncan’s new multiple range test (*p* < 0.05).

## Data Availability

The data presented in this study are available on request from the corresponding author. The data are not publicly available as they are part of ongoing research.

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
