# Peer review of "Formulation and Evaluation of Chitosan/NaCl/Maltodextrin Microparticles as a Saltiness Enhancer: Study on the Optimization of Excipients for the Spray-Drying Process"

_polymers, 2021, doi:10.3390/polym13244302_

Round 1

Reviewer 1 Report

Manuscript "Formulation and Evaluation of Chitosan / NaCl Microparticles as a Saltiness Enhancer: Study on the Optimization of Excipients for the Spray Drying Process" presents an interesting and innovative topic. The research used many methods to compare the tested samples, the research is well planned. The discussion of the results is correct.

Detailed comments:

line 130 - why were converted into pullulans?

Figure 2 - the charts are very unreadable, is it not possible to present all the results on a larger chart? 

Reviewer 2 Report

The article presents the possibility of obtaining spray-dried powders (Chitosan/NaCl/Maltodextrin formulations) ensuring better saltiness perception than native NaCl. The article contains the physicochemical properties of the formulations obtained. The article is interesting, but it has some weaknesses (shown below):

Title and elsewhere. All formulations contain maltodextrin so the title should be modified. “Chitosan/NaCl/ Maltodextrin Microparticles” not “Chitosan/NaCl Microparticles”.

L342-345. Spray dried maltodextrin is an amorphous state. It does not have a crystalline region. Improve.

L375-376. I cannot agree with that. Maltodextrin-based powders are characterized by increased hygroscopicity. In the previous article by the authors [19] hygroscopic rate was lower (the most similar systems: CsAN10, CsAN20).

Table 5. Please add statistic for caking strength (last column).

L435-437. In this study the authors did not present various process conditions. The statement “best handling properties could be obtained at 150 °C of inlet temperature” is unfounded. How do you know that a temperature of 175 or 160 ° C is not optimal?

L444-446. “Adding chitosan in the spray drying formulation resulted in heterogeneous microparticles…”? All formulations contained chitosan; the authors did not show the matrix structure without chitosan.

Disscusion

The discussion lacks a comparison of the obtained results in relation to the physical properties of the main components, salt or maltodextrin.

The authors have made no reference to the low efficiency of the process, which affects their application in industry.

Conclusions

The obtained formulations seem problematic during their turnover (transport, dosing, storage) due to small particle size, dustiness, poor flowability, and hygroscopicity. Of the tested formulations, MCC15 was the best, but its use seems to be still very limited and unprofitable (low efficiency). Is MCC15 formula worthy of further investigation? Moreover, in many countries the use of chitosan as a food additive is not allowed.

Furthermore, bulk materials, such as food powders, often gain strength and cake during storage and handling, whereby free flowing materials are transformed into lumps or a total caked bulk. This can be very problematic, whereby the product may be rendered unusable due to severe caking. Besides, during handling, storage, processing and distribution to the final consumer, the powders may experience variations in temperatures and atmospheric humidities, which may alter the handling behaviour and appearance of the powders. This is especially important if powders are transported to hotter, more humid climates, where a mix may cake solidly or liquefy from sorbing water.

Round 2

Reviewer 2 Report

I have no comments. The manuscript has been sufficiently improved.